# Query-as-context Pre-training for Dense Passage Retrieval

**Xing Wu**[1,2*] **Guangyuan Ma**[1,2*] **Wanhui Qian**[3]
**Zijia Lin**[4], and **Songlin Hu**[1,2†]
[1] Institute of Information Engineering, Chinese Academy of Sciences
[2] School of Cyber Security, University of Chinese Academy of Sciences
[3] Du Xiaoman Financial, [4] Kuaishou Technology
{wuxing, maguangyuan, husonglin}@iie.ac.cn, qianwanhui@duxiaoman.com
linzijia07@tsinghua.org.cn

## Abstract

Recently, methods have been developed to improve the performance of dense passage retrieval by using context-supervised pre-training. These methods simply consider two passages from the same document to be relevant, without taking into account the potential negative impacts of weakly correlated pairs. Thus, this paper proposes *query-as-context* pre-training, a simple yet effective pre-training technique to alleviate the issue. Query-as-context pre-training assumes that the query derived from a passage is more likely to be relevant to that passage and forms a passage-query pair. These passage-query pairs are then used in contrastive or generative context-supervised pre-training. The pre-trained models are evaluated on large-scale passage retrieval benchmarks and out-of-domain zero-shot benchmarks. Experimental results show that query-as-context pre-training brings considerable gains for retrieval performances, demonstrating its effectiveness and efficiency.

## 1 Introduction

Passage retrieval is the process of retrieving relevant passages from a large corpus in response to a query, which is useful in a variety of downstream applications such as web search (Fan et al., 2021; Guo et al., 2022; Lin et al., 2021a), question answering (Karpukhin et al., 2020; Lee et al., 2020; Zhu et al., 2021) and dialogue systems (Gao et al., 2022a; Yu et al., 2021). The success of pre-trained language models (PLMs) (Devlin et al., 2018; Liu et al., 2019) has led to the development of more powerful PLM-based dense and sparse passage retrieval approaches.

PLM-based dense retrieval methods (Xiong et al., 2020; Lu et al., 2021; Hofstätter et al., 2021; Gao and Callan, 2021b; Ren et al., 2021b; Ma et al., 2022; Liu and Shao, 2022; Wu et al., 2022;

**Passage 1**. A good day at Pipeline means an encounter with fellow surfers who can be as friendly as pitbulls with migraines, and waves that can shatter boards into kindling. And then there's the reef. At Pipeline there can be 10 foot waves blasting over just three feet of water, so if you fall on the reef or get caught inside the break, you're lucky if you come out merely sliced up.

**Passage 2**. This idyll can still be found on the island of Lanai. Formerly a pineapple plantation, Lanai is almost entirely owned by billionaire entrepreneur David Murdock, and its only notable commercial hub is a tiny village built around a square of Cook pine trees.

Figure 1: An example of low-relevance passages within a document from the MS-MARCO corpus. The two passages are weakly correlated in content.

Wang et al., 2022) use PLMs to encode queries and passages into a shared semantic space. The semantic relationships between query and passage representations are then measured by dot product or cosine similarities. Pre-training and fine-tuning techniques have been developed to improve the performance of dense retrieval models. Pre-training processes for dense retrieval aim to improve the text representation modeling ability of the encoder through auxiliary self-supervised or context-supervised tasks.

Context-supervised pre-training (Gao and Callan, 2021b; Wu et al., 2022) assumes that two passages[1] within the same document are contextual or related to each other and can therefore be used for contrastive learning or contextual decoding. However, context-supervised pre-training ignores the fact that the passages within a document may be weakly related or even irrelevant in many cases. As shown in Figure 1, two passages within a document from the MS-MARCO corpus (Nguyen et al., 2016) are not directly related

---

*The first two authors contributed equally to this work.
†Corresponding authors.

[1]A passage refers to a long text span consisting of consecutive sentences within a much longer document.

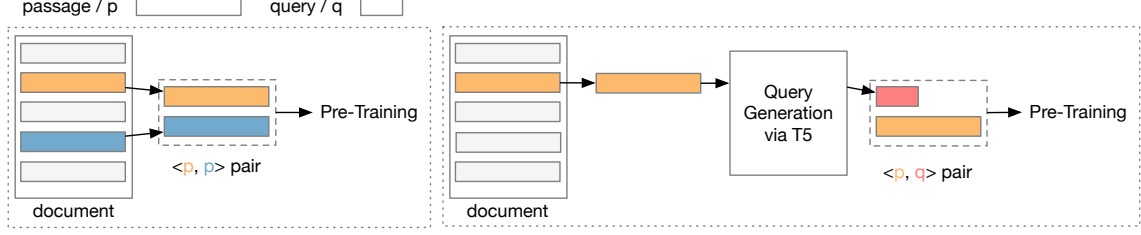

Figure 2: A comparison of context-supervised pre-training and query-as-context pre-training.

in content. According to our statistic results via human annotation, only 35.5% of passage pairs in the training data of coCondenser (Gao and Callan, 2022) have high correlation. For statistical details, please refer to Appendix A. These weakly correlated or irrelevant passages do not align with the assumptions on which context-supervised pre-training is based, and are likely to be detrimental to context-supervised pre-training.

In contrast to dense retrieval, sparse retrieval is based on the "bag-of-words" assumption and represents passages and queries as sparse term-based vectors. PLM-based sparse retrieval (Nogueira and Lin, 2019; Dai and Callan, 2019; Mao et al., 2020; Bai et al., 2020; Formal et al., 2021b,a; Mallia et al., 2021; Shen et al., 2022) uses PLM to improve sparse vectors. One representative technique is Query Prediction (Nogueira and Lin, 2019), which predicts a set of relevant queries to enrich the passage's content and thus alleviates the mismatch problem. Query prediction has been shown to be effective in sparse retrieval, but has not yet been explored in the context of dense retrieval, especially in the pre-training process. This raises the question of whether the query prediction technique can benefit the pre-training process tailored for dense retrieval.

The observation that predicted queries align better with the content of a passage in our statistical analyses (see Appendix A) suggests that query prediction could be a promising way to alleviate the issue of weakly correlated passages for context-supervised pre-training. Thus, this paper focuses on exploring query prediction techniques to improve context-supervised pre-training methods for dense retrieval. Our proposed method, termed *query-as-context* pre-training, assumes that a query derived from a passage (using a generative model like T5) is more likely to be a **relevant context** to the passage. In contrast to the previous context-supervised methods that create a training pair using

two randomly selected passages from a document, the query-as-context method generates a training pair by combining a passage with a predicted query, as illustrated in Figure 2.

There are several advantages to using the query-as-context setting. Firstly, the query is more likely to be related to the passage because it is generated from the passage. Additionally, the use of passage-query pairs in supervised downstream retrieval training is consistent with using passage-query pairs in pre-training, which helps to bridge the gap between the two processes. Finally, since the passage-query pairs are generally shorter than the previously used passage-passage pairs, it speeds up the pre-training process and reduces the training overhead.

To verify the effectiveness of our proposed query-as-context pre-training, we conduct experiments on large-scale web search benchmarks: MS-MARCO Passage Ranking (Nguyen et al., 2016), TREC Deep Learning (DL) Track 2019 (Craswell et al., 2020a) and Track 2020 (Craswell et al., 2020b). We also evaluate query-as-context pre-trained models on the BEIR (Thakur et al., 2021) benchmark with a large set of out-of-domain datasets. Experimental results show that query-as-context achieves considerable gains over competing baselines.

Our contributions can be summarized as follows:

- We reveal the previously ignored issue of weakly correlated passage pairs during context-supervised pretraining.

- We propose query-as-context pre-training, a simple yet effective pre-training technique to alleviate the issue above.

- Experiments show that query-as-context pre-training brings considerable gains and meanwhile speeds up pre-training.

## 2 Preliminary: Context-supervised Pre-training

In this section, we begin by providing an overview of the pre-training corpus. Subsequently, we describe the masked language modeling task, which serves as a foundational task of pre-training. Finally, we present two representative contrastive and generative context-supervised pre-training methods, on which our proposed query-as-context will be applied.

**Pre-training Corpus**  Given a set of documents, we randomly extract pairs of passages from each document, which forms a training corpus as follows:

$$\{\{\mathbf{x}_0, \mathbf{y}_0\}, ..., \{\mathbf{x}_m, \mathbf{y}_m\}\} \quad (1)$$

where $\{\mathbf{x}_i, \mathbf{y}_i\}$ is a pair of passages from the same document.

**Masked Language Modeling (MLM)**  Formally, given a passage $\mathbf{x}$ with $n$ tokens, a special token [CLS] is added to the beginning of the passage, resulting in

$$\mathbf{x} = \{x_0, x_1, ..., x_n\} \quad (2)$$

where $x_0$ represents the [CLS] token. Then, a certain percentage of positions are randomly selected as "mask positions" ($mask\_pos$) and are replaced with a special token [MASK] or a random token. The masked passage is then passed through a text encoder, which commonly consists of $L$ layers of transformer blocks. For the $l$-th transformer layer in the encoder, its outputs are the hidden states of the layer

$$\mathbf{h}^l = \{h_0^l, h_1^l, ..., h_n^l\} \quad (3)$$

The output of the last layer is then used to calculate the MLM's target loss

$$\mathcal{L}_{mlm} = - \sum_{i \in mask\_pos} CE(\phi(h_i^L), x_i) \quad (4)$$

where $CE$ is short for cross entropy function and $\phi$ is a projection of the corresponding hidden states of $x_i$ to a vocabulary distribution.

### 2.1 coCondenser

coCondenser (Gao and Callan, 2021b) is a representative *contrastive* context-supervised method. For coCondenser, two passages from a document are considered relevant and form a positive pair, while two passages from different documents are

considered as irrelevant and form a negative pair. These pairs constitute mini-batches for contrastive learning. A common approach for generating an embedding representation of a passage is to use the hidden states of the [CLS] position in the last layer of the encoder, i.e., $h_0^L$. Thus, the embedding representations of passages $\mathbf{x}$ and $\mathbf{y}$ are $h_0^L(\mathbf{x})$ and $h_0^L(\mathbf{y})$, simplified as $h_\mathbf{x}$ and $h_\mathbf{y}$. Then, for a mini-batch $\mathbf{B}$, the contrastive learning objective w.r.t $\mathbf{x}$ is formulated as:

$$\mathcal{L}_{co} = - \log \frac{\exp(\text{sim}(h_\mathbf{x}, h_\mathbf{y})/\tau)}{\sum_{h' \in \mathbf{B}} \exp(\text{sim}(h_\mathbf{x}, h')/\tau)} \quad (5)$$

where $\tau$ is a temperature hyper-parameter and $\text{sim}(,)$ is the dot product similarity function.

An additional auxiliary decoder is also appended to the encoder, which consists of $N$ layers of transformers. The auxiliary decoder takes the concatenation of the [CLS] representation from the $L$-th layer, i.e., $h_0^L$, and the token representations from the encoder's $M$-th (e.g. 6-th) layer, i.e., $\{h_1^M, ..., h_n^M\}$, as inputs. Similar to MLM, the output of the auxiliary decoder's last layer is then used to perform an auxiliary MLM pre-training.

$$\mathcal{L}_{mlm}^{aux} = - \sum_{i \in mask\_pos} CE(\phi(h_i^N), x_i) \quad (6)$$

Finally, the total loss of coCondenser is:

$$\mathcal{L} = \mathcal{L}_{mlm} + \mathcal{L}_{mlm}^{aux} + \mathcal{L}_{co} \quad (7)$$

For more details, please refer to (Gao and Callan, 2021b).

### 2.2 CoT-MAE

CoT-MAE (Wu et al., 2022) is a representative *generative* context-supervised method that uses an asymmetric encoder-decoder structure, with a deep encoder of $L$ layers and a shallow decoder of $N$ layers. It performs MLM training on both the encoder and the decoder simultaneously. For a pair of passages $\{\mathbf{x}, \mathbf{y}\}$, suppose $\mathbf{x}$ is fed into the encoder side and $\mathbf{y}$ is fed into the decoder side.

On the encoder side, $\mathbf{x}$ is reconstructed using only the unmasked tokens in the passage, similar to BERT's MLM process, but with a higher mask rate (e.g. 30%). On the decoder side, $\mathbf{y}$ is reconstructed using both its unmasked tokens and the contextual passage $\mathbf{x}$. The decoder takes the sentence embedding of $\mathbf{x}$, i.e., $h_\mathbf{x}$, and the word representations of masked $\mathbf{y}$ as input, which are concatenated as:

$$\mathbf{d}^0 = \{h_\mathbf{x}, y_1, ..., y_n\} \quad (8)$$

The concatenation $\mathbf{d}^0$ is then passed through the $N$ layers of Transformer blocks, and the hidden states of $k$ layer is formulated as:

$$\mathbf{d}^k = \{d_0^k, d_1^k, ..., d_n^k\} \quad (9)$$

The outputs of the last layer in decoder are then used for LM pre-training, with the loss defined as:

$$\mathcal{L}_{ctx\_mlm} = -\sum_{i \in mask\_pos} CE(\phi(d_i^N), y_i) \quad (10)$$

The subscript $ctx$ denotes the process is context-supervised. Then, we add the losses from both the encoder and the decoder to get the final loss:

$$\mathcal{L} = \mathcal{L}_{mlm} + \mathcal{L}_{ctx\_mlm} \quad (11)$$

For more details, please refer to (Wu et al., 2022).

# 3 Query-as-context Pre-training

In this section, we first introduce the details of query-as-context pre-training, and then introduce the fine-tuning process of the pre-trained models on the retrieval tasks.

## 3.1 Pre-training

Pre-training is conducted on a large scale of documents without annotations. For each document $\mathbf{D}$, we extract a set of passages with a maximum length, $\{\mathbf{x}_0, \mathbf{x}_1, ...\}$. Following (Nogueira and Lin, 2019), for each passage $\mathbf{x}_i$, we use a fine-tuned T5 model for generating queries. We apply nucleus sampling with $\mathbf{top_p}$=0.95 and $\mathbf{top_k}$=25 to produce multiple queries for promoting diversity.

Specially, each passage $\mathbf{x}_i$ will be fed into the fine-tuned T5 model, and generate $C$ candidate queries, $\{\mathbf{q}_{ij}\}_{j=1}^C$. During training, we will randomly select one of the candidate queries to use as the context for the passage:

$$\mathbf{y_i} = sample(\{\mathbf{q}_{ij}\}_{j=1}^C)$$

The passage and sampled query form a training pair $\{\mathbf{x}_i, \mathbf{y}_i\}$, which can be used to replace the original pair used in Equation 1. Specifically, the passage-query pair are directly used for contrastive pre-training of coCondenser. For CoT-MAE, we fed the passage into the encoder, and query into the decoder for generative pre-training. Model implementations for coCondenser and CoT-MAE have been introduced in Section 2.1 and 2.2.

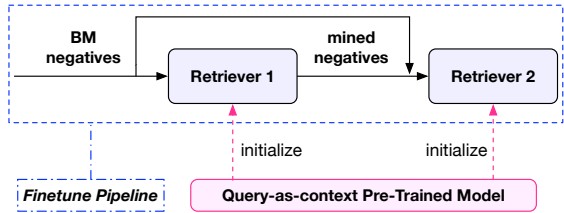

Figure 3: Illustration of the fine-tuning pipeline. The query-as-context pre-trained model is used to initialize the dual-encoder retrievers.

## 3.2 Fine-tuning

We fine-tune on the downstream retrieval tasks to verify the effectiveness of pre-training. Following (Gao and Callan, 2021b; Wu et al., 2022), the fine-tuning process on MS-MARCO is based on a two-stage pipeline with hard negative mining (Gao et al., 2022b), as depicted in Figure 3. This process involves two stages of training, bi-encoder retriever 1 and bi-encoder retriever 2, which are both initialized with the query-as-context pre-trained models. The retrievers are trained with contrastive learning on the manually annotated passage-query pairs. For a manually annotated passage-query pair $(p^+, q^+)$, the representations of the passage and the query form a positive example $(h_{p^+}, h_{q^+})$. When training retriever 1, for query $q^+$, the negative samples $\{p^-\}$ include in-batch negative passages and BM25 mined hard negative passages. When training retriever 2, hard negatives are also mined using a well-trained retriever 1 and combined with the other negative passages to create the negative samples $\{p^-\}$. Both stages are optimized using the InfoNCE loss.

$$\mathcal{L}^{ft} = -\log \frac{\exp(\mathrm{sim}\left(h_{q^+}, h_{p^+}\right)/\tau)}{\sum\limits_{p \in \{p^+, p^-\}} \exp(\mathrm{sim}\left(h_{q^+}, h_p\right)/\tau)} \quad (12)$$

where $\tau$ is a temperature hyper-parameter fixed to 1 and $\mathrm{sim}\,(,)$ is dot product similarity function.

Following (Thakur et al., 2021), we train the retriever with MS-MARCO negatives[2] for the out-of-domain evaluation on BEIR benchmarks

# 4 Experiment

In this section, we provide details on the pre-training and fine-tuning processes. Then we present the experimental results.

---

[2] https://sbert.net/datasets/ msmarco-hard-negatives.jsonl.gz

## 4.1 Pre-training

**Query-as-context Dataset**  Following (Gao and Callan, 2021b; Wu et al., 2022), the pre-training dataset is collected from the MS-MARCO passages corpus, which contains 3.2 million documents. We use NLTK to split each document into sentences, and group these sentences into passages of no more than 144 consecutive tokens. For each passage, we generate candidate queries via a public T5 model [3]. During pre-training, we select a batch of passages at each step and randomly choose a candidate **query as context** for each passage to form a relevant pair.

**Model Implementation**  Following (Wu et al., 2022), the encoder for **CoT-MAE** is initialized with a pre-trained 12-layer BERT-base model, while the decoder is initialized from scratch. We pre-train the model using the AdamW optimizer for a maximum of 50k steps, with a learning rate of 4e-4, a batch size of 16k, and a linear schedule with a warmup ratio of 0.1. We use 16 Tesla V100 GPUs to train the model for 60 hours, and then discard the decoder, leaving only the encoder for fine-tuning. Following (Gao and Callan, 2021b), the encoder for **coCondenser** is initialized from a pre-trained 12-layer Condenser (Gao and Callan, 2021a) model. The training is conducted on 8 Tesla V100 GPUs for 120,000 steps over 90 hours using the AdamW optimizer with a learning rate of 1e-4, a global batch size of 2k, and a weight decay of 0.01. Once the pre-training is finished, the Condenser head is discarded, resulting in a model with the same architecture as $\text{BERT}_{base}$ for fine-tuning.

## 4.2 Fine-tuning

**Datasets and Evaluation**  We fine-tune the pre-trained coCondenser and CoT-MAE on MS-MARCO passage ranking (Nguyen et al., 2016), TREC Deep Learning (DL) Track 2019 (Craswell et al., 2020a) and 2020 (Craswell et al., 2020b) tasks for evaluation.

MS-MARCO (Nguyen et al., 2016) is a benchmark dataset that contains real user queries collected from Bing search and web pages, and includes approximately 8.8 million passages in total. The training set consists of around 500,000 annotated query-document pairs, while the dev set contains 6,980 annotated queries. Since the test set is not publicly available, the dev set is used

[3] https://huggingface.co/doc2query/all-with_prefix-t5-base-v1

for evaluation following previous work (Gao and Callan, 2021b; Wu et al., 2022). We evaluate our performance on MS-MARCO using MRR@10, Recall@50, and Recall@1K.

TREC Deep Learning (DL) (Craswell et al., 2020a,b) tracks provide test sets with more elaborate annotations to evaluate the real capacity of ranking models. We evaluate the 2019 and 2020 test sets. The 2019 test set contains 43 annotated queries and the 2020 test set contains 54 annotated queries. We evaluate our performance on TREC with NDCG@10.

**Implementation**  We reuse a widely adopted evaluation pipeline (Gao and Callan, 2021b; Wu et al., 2022; Gao et al., 2022b), with a common random seed (42) to support reproducibility. Note that, as we focus on improving the pre-training technique, we do NOT use any enhanced methods, such as distillation from a strong re-ranker (Ren et al., 2021b; Santhanam et al., 2021) or multi-vector representation (Khattab and Zaharia, 2020), which can lead to further improvements. The fine-tuning is only trained on the MS-MARCO dataset and evaluated on the dev set and TREC DL 2019/2020 test sets. It's trained on 8 Tesla V100 GPUs using the AdamW optimizer with a learning rate of 2e-5, a global batch size of 64, and a weight decay of 0.01. The passage length is also set to 144, the negative depth is set to 200 and the number of negative passages for one query in the fine-tuning iteration is 15.

## 4.3 Baselines

Our baseline methods include the sparse retrieval method and the dense retrieval method, as shown in Table 1. Results of sparse retrieval baselines are mainly from (Qu et al., 2020), including BM25, docT5query (Nogueira and Lin, 2019), DeepCT (Dai and Callan, 2019) and GAR (Mao et al., 2020). Results of dense retrieval baselines are mainly from (Gao and Callan, 2021b; Liu and Shao, 2022; Ren et al., 2021b; Ma et al., 2022), including ANCE (Xiong et al., 2020), SEED (Lu et al., 2021), TAS-B (Hofstätter et al., 2021), RetroMAE (Liu and Shao, 2022), SimLM (Wang et al., 2022) and etc. We compare the query-as-context performances with their baselines on both retriever 1 and retriever 2.

## 4.4 Main Results

As shown in Table 1, the results demonstrate that query-as-context pre-training leads to improved

| Model | MS-MARCO | | | TREC DL 19 | TREC DL 20 |
| | MRR@10 | R@50 | R@1k | NDCG@10 | NDCG@10 |
|---|---|---|---|---|---|
| **Sparse retrieval** | | | | | |
| BM25 | 18.7 | 59.2 | 85.7 | 51.2 | 47.7 |
| DeepCT (Dai and Callan, 2019) | 24.3 | 69.0 | 91.0 | 57.2 | - |
| docT5query (Nogueira and Lin, 2019) | 21.5 | 64.4 | 89.1 | 64.2 | - |
| **Dense retrieval** | | | | | |
| NPRINC (Lu et al., 2020) | 31.1 | - | 97.7 | - | - |
| ANCE (Xiong et al., 2020) | 33.0 | - | 95.9 | 64.5 | 64.6 |
| SEED (Lu et al., 2021) | 33.9 | - | 96.1 | - | - |
| TAS-B (Hofstätter et al., 2021) | 34.0 | - | 97.5 | 71.2 | 69.3 |
| COIL (Gao et al., 2021) | 35.5 | - | 96.3 | 70.4 | - |
| ColBERT (Khattab and Zaharia, 2020) | 36.0 | 82.9 | 96.8 | - | - |
| COSTA (Ma et al., 2022) | 36.6 | 84.1 | 97.3 | - | 67.8 |
| Condenser (Gao and Callan, 2021a) | 36.6 | - | 97.4 | 69.8 | 66.5 |
| RocketQA (Qu et al., 2020) | 37.0 | 85.5 | 97.9 | - | - |
| PAIR (Ren et al., 2021a) | 37.9 | 86.4 | 98.2 | - | - |
| SimLM (Wang et al., 2022) | 39.1 | - | 98.6 | 71.4 | 69.7 |
| RetroMAE (Liu and Shao, 2022) | 39.3 | - | 98.5 | 68.1 | 70.6 |
| LED (Zhang et al., 2022a) | 39.6 | 86.6 | 98.3 | 70.5 | 67.9 |
| coCondenser (Gao and Callan, 2021b) | 38.2 | 86.5 | 98.4 | 71.7 | 68.4 |
| coCondenser (120K) - retriever 1 † | 37.0 | 86.0 | 98.5 | 68.2 | 68.8 |
| w/ query-as-context (120K) - retriever 1 | **37.4** | **87.3** | **98.6** | 68.1 | 69.2 |
| coCondenser (120K) - retriever 2 † | 38.8 | 87.8 | 98.8 | 71.1 | 68.4 |
| w/ query-as-context (120K) - retriever 2 | **39.4** | **88.6** | **99.0** | **73.1** | **71.8** |
| CoT-MAE (Wu et al., 2022) | 39.4 | 87.0 | 98.7 | 70.9 | 70.4 |
| CoT-MAE (50K) - retriever 1† | 37.2 | 85.7 | 98.2 | 65.7 | 66.5 |
| w/ query-as-context (50K) - retriever 1 | **38.6** | **87.7** | **98.6** | 67.7 | 67.8 |
| CoT-MAE (50K) - retriever 2† | 38.8 | 87.3 | 98.6 | 70.7 | 69.7 |
| w/ query-as-context (50K) - retriever 2 | **40.2** | **88.8** | **98.8** | **71.5** | **72.7** |

Table 1: Main results on MS-MARCO passage ranking and TREC DL datasets. † denotes our reproduction using publicly available codes. The score that is better in comparison is marked in **bold**.

performance.

**coCondenser** When reproducing coCondenser, the pre-training steps extend to 120k steps. The main evaluation metric, MRR@10 on the MS-MARCO passage ranking dataset, of retriever 2 improves by 0.6pp compared to the original paper(Gao and Callan, 2021b). When query-as-context pre-training is used, there is a further improvement of 0.6pp on MRR@10. On both TREC DL 19 and 20 test sets, there are improvements of 2pp on DL 19 and 3.4pp on DL 20. In addition, query-as-context pre-training also improves the MRR@10 and R@50 scores of retriever 1.

**CoT-MAE** When reproducing CoT-MAE, for efficiency, we adopt a much larger batch size than in (Wu et al., 2022), which allows us to reduce the number of training steps from 1200k to 50k. This results in faster training, but somehow lower performance on the MS-MARCO MRR@10 metric com-

pared to the original paper. However, when query-as-context pre-training is applied, there is an obvious improvement of 1.4pp on MRR@10, reaching 40.2. Even compared to the 1200k model's performance in the original paper, we still achieve a non-trivial improvement of 0.8pp. To the best of our knowledge, this is the new state-of-the-art result for a single vector pre-trained (not a reranker-distilled) dense retriever. On both TREC DL 19 and 20 test sets, there are improvements of 0.8pp on DL 19 and 3pp on DL 20. In addition, query-as-context pre-training also improves the MRR@10, R@50, and R@1k scores of retriever 1.

Overall, the query-as-context pre-training approach is effective, improving both contrastive and generative context-supervised pre-training. This is due to two main reasons: (1) Pre-trained models can provide better parameters initialization for both retriever 1 and retriever 2; (2) A better retriever 1 can be used to mine more effective hard negatives,

| Dataset | coCondenser | | CoT-MAE | |
|---|---|---|---|---|
| | w/o | w/ | w/o | w/ |
| trec-covid | 0.632 | **0.703** | 0.646 | **0.665** |
| nfcorpus | **0.333** | 0.330 | 0.319 | **0.340** |
| nq | 0.531 | **0.548** | 0.513 | **0.546** |
| hotpotqa | 0.538 | **0.583** | 0.512 | **0.572** |
| fiqa | 0.319 | **0.322** | 0.288 | **0.326** |
| arguana | 0.389 | **0.447** | 0.312 | **0.416** |
| webis-touche2020 | **0.213** | 0.204 | 0.202 | **0.212** |
| cqadupstack | 0.310 | **0.341** | 0.312 | **0.337** |
| quora | **0.866** | 0.864 | 0.781 | **0.859** |
| dbpedia-entity | 0.373 | **0.386** | 0.355 | **0.406** |
| scidocs | 0.133 | **0.145** | 0.132 | **0.151** |
| fever | **0.728** | 0.664 | **0.707** | 0.688 |
| climate-fever | **0.204** | 0.199 | 0.173 | **0.220** |
| scifact | 0.599 | **0.648** | 0.591 | **0.642** |
| Average | 0.441 | **0.456** | 0.417 | **0.456** |

Table 2: Out-of-domain evaluation on BEIR benchmark. The score that is better in comparison is marked in **bold**.

which further improves the training of retriever 2.

### 4.5 Out-of-domain Evaluation

We evaluate the out-of-domain performance of query-as-context pre-trained models on the zero-shot benchmark BEIR(Thakur et al., 2021). BEIR benchmark contains 9 different open-domain information retrieval tasks from 18 different datasets. We evaluate the models on the 14 publicly available datasets[4]. As shown in the table, both the coCondenser and the CoT-MAE results show non-trivial improvements on most datasets when using query-as-context pre-training. Specifically, using query-as-context pre-training improves the performance of the coCondenser model on 9 different datasets. The improvement in CoT-MAE is more significant, with notable gains observed on 13 datasets.

### 5 Analyses

In this section, we examine the efficiency advantage and analyze the impact of different settings on query-as-context pre-training.

### 5.1 Impact of Generated Query Number

During pre-training, using multiple candidate queries leads to better diversity as each passage is paired with a different candidate query in each epoch. Therefore, we explore the effect of the number of generated queries. As shown in Table 3,

---

[4]The current state-of-the-art models on the BEIR benchmark reach higher scores as they are pre-trained on the WIKI dataset. Due to the high cost of pre-training, we directly evaluate the models pre-trained on the MS-MARCO dataset and leave the exploration on the WIKI dataset in future work.

for coCondenser, increasing the number of queries from 1 to 5 slightly improves performance on the MS-MARCO dataset and leads to a good improvement on the TREC DL 19 and 20 test sets. For CoT-MAE, using 5 queries lead to an increase on the MS-MARCO dataset and TREC DL 20 test set, while a slight performance decrease in the TREC DL 19 test set. However, further increasing the number of candidate queries will generally bring about a decline in performance. A proper number of queries retains their correlation to the passages, thus yielding higher performance in query-as-context pre-training.

### 5.2 Impact of Mixed Context

We further explore the effect of mixing the two kinds of contextual pairs, passage-query and passage-passage. In a training step, we randomly choose to use either the passage-query or passage-passage pair as input with the same probability. As shown in Table 4, mixing does not improve the effect for coCondenser and CoT-MAE, despite increasing the diversity of context. The decrease aligns with the human-annotated correlation results in Appendix A. The passage-passage pairs have a higher proportion of low correlation pairs, so combining passage-query and passage-passage pairs will be less effective than using passage-query pairs alone. This also indicates that for pre-training tailored for intensive retrieval, the relevance of training pairs is more crucial than diversity.

### 6 Related Works

**Dense Retrieval**  Different techniques have been developed to improve dense retrieval, both in fine-tuning and pre-training stages. In fine-tuning stage, attempts includes mining hard negatives (Xiong et al., 2020; Zhan et al., 2021), late interaction (Khattab and Zaharia, 2020), query clustering (Hofstätter et al., 2021), reranker distillation (Lin et al., 2021b; Santhanam et al., 2021), data augmentation (Qu et al., 2020) and jointly learning (Ren et al., 2021b; Zhang et al., 2022b, 2021) . In pre-training stages, attempts are divided into two categories. One category focuses on improving the encoder using auxiliary self-supervised auto-encoding tasks (Lu et al., 2021; Gao and Callan, 2021a; Liu and Shao, 2022; Zhou et al., 2022). The other category proposes passage prediction tasks to resemble passage retrieval in pre-training (Chang et al., 2020; Gao and Callan, 2021b; Ma et al., 2022).

| Model | Query Number | MS-MARCO | | | | | | TREC DL 19 NDCG@10 | TREC DL 20 NDCG@10 |
| | | Retriever-1 | | | Retriever-2 | | | | |
| | | MRR@10 | R@50 | R@1k | MRR@10 | R@50 | R@1k | | |
|---|---|---|---|---|---|---|---|---|---|
| coCondenser | 1 | 37.7 | **87.6** | 98.6 | 39.3 | 88.5 | 98.9 | 72.3 | 71.1 |
| | 5 | 37.4 | 87.3 | **98.6** | **39.4** | **88.6** | **99.0** | **73.1** | **71.8** |
| | 10 | 37.6 | 87.0 | 98.5 | 39.1 | 88.5 | 98.9 | 71.1 | 70.9 |
| | 20 | **37.7** | 87.2 | 98.6 | 39.4 | 88.3 | 99.0 | 71.5 | 70.3 |
| CoT-MAE | 1 | 38.3 | 87.4 | 98.5 | 39.9 | 88.7 | 98.7 | 71.7 | 70.8 |
| | 5 | **38.6** | **87.7** | **98.6** | **40.2** | **88.8** | 98.8 | 71.5 | **72.7** |
| | 10 | 38.5 | 87.2 | 98.6 | 39.7 | 88.7 | 98.8 | **72.5** | 71.7 |
| | 20 | 38.3 | 87.5 | 98.6 | 39.7 | 88.5 | 98.8 | 72.2 | 69.9 |

Table 3: Impact of the number of generated queries. The score that is better in comparison is marked in **bold**.

| Model | Mixed | MS-MARCO | | | | | | TREC DL 19 NDCG@10 | TREC DL 20 NDCG@10 |
| | | Retriever-1 | | | Retriever-2 | | | | |
| | | MRR@10 | R@50 | R@1k | MRR@10 | R@50 | R@1k | | |
|---|---|---|---|---|---|---|---|---|---|
| coCondenser | ✗ | **37.4** | **87.3** | **98.6** | **39.4** | **88.6** | **99.0** | **73.1** | **71.8** |
| | ✓ | 37.4 | 86.7 | 98.4 | 39.1 | 88.1 | 98.8 | 71.2 | 71.3 |
| CoT-MAE | ✗ | **38.6** | **87.7** | **98.6** | **40.2** | **88.8** | **98.8** | 71.5 | **72.7** |
| | ✓ | 36.9 | 85.6 | 98.1 | 38.4 | 87.1 | 98.5 | **72.4** | 70.0 |

Table 4: Effect of mixing passage-query and passage-passage pairs in pre-training.

The most related methods in this category are (Gao and Callan, 2021b) and (Wu et al., 2022). (Gao and Callan, 2021b) introduces a context-supervised contrastive pre-training process, with the hypothesis that passages from the same document are closer than those from different documents. (Wu et al., 2022) introduces a context-supervised generative masked auto-encoding task via the decoder-side reconstruction task assisted by contextual embedding. Our work is on the basis of these two methods.

**Query Prediction** Query Prediction is a technique originally introduced to the IR community to expand passages. It can significantly improve the performance of BM25 by generating additional queries and appending them to passages before building the inverted index (Nogueira and Lin, 2019). Query prediction has also been used to learn better sparse (Mallia et al., 2021) or dense (Li et al., 2022) representations for documents. In scenarios where data is scarce, query prediction can be used for domain adaptation by generating synthetic queries on target domains for model training (Ma et al., 2020). To reduce noise in the generated data, a cross-encoder can also be used for pseudo-labeling (Wang et al., 2021). The most related work to ours is (Li et al., 2022), which encodes each document with a set of generated pseudo-queries to obtain query-informed document representations. However, (Li et al., 2022) focuses on improving the fine-tuning process for dense retrieval, while we are working on the pre-training process.

## 7 Conclusions

In this work, we propose query-as-context pre-training, a simple yet effective technique to alleviate the previously ignored issue of weakly correlated pairs during context-supervised pre-training. Extensive experiments well validate its effectiveness and efficiency.

## 8 Limitations

A passage is more likely to have a high correlation with its corresponding generated query than another randomly selected passage from the same document. However, limited by the capabilities of the T5 model, there are still a large number of unrelated passage-query pairs. We believe that more powerful large language models have the potential to further alleviate this problem, which is left to our future research.

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

## A  Statistically Analysis of Weakly Correlated Passages

We randomly select 200 documents from the MS-MARCO dataset and randomly select a passage from each document. Then we construct the contextual pairs in two ways:

1. **Random passage-passage pair**: Referring to coCondenser (Gao and Callan, 2022), we randomly select another passage within the same document as the context for the passage.

2. **Generated passage-query pair**: Referring to the out-of-shelve docT5query (Nogueira and Lin, 2019), we use query prediction technology to generate a query as the context for the passage.

We asked the annotators to label whether the random contexts or generated queries are strongly related to the corresponding passages. We manually annotate the 200 passage-passage pairs and passage-query pairs as high-correlation or low-correlation respectively. To eliminate preference bias, we divide 6 annotators into two groups. One group annotates 100 passage-passage pairs and 100 passage-query pairs, while the other annotates the remaining pairs. The correlation of each pair is voted by the annotation results of three annotators. The statistical results are shown in Table 5.

Only 35.5% of the passage-passage pairs are highly correlated, compared to 56.6% of the passage-query pairs. Therefore, we suggest that the generated query is a more relevant context than the randomly sampled passages. However, due to the limited ability of the base-sized T5 model, nearly half of the generated queries are still not quite exact or strongly correlate to the corresponding passage. We will further explore the potential ability to utilize large language models to generate more precise and semantic correlate queries for improving the performance boundaries of dense passage retrieval pre-training.

| Pairs | Random passage-passage | Generated passage-query |
|---|---|---|
| **Correlation rate** | 35.5% | **56.5%** |

Table 5: Correlation statistics of human annotation results of different contextual pairs, each with 200 pairs. The score that is better in comparison is marked in **bold**.