# OpenReview forum: "Query-as-context Pre-training for Dense Passage Retrieval"
_EMNLP/2023/Conference — EMNLP 2023 Main_

### Official Review · Reviewer_qcxv · 2023-07-31

**Soundness:** 4

**Excitement:**

3: Ambivalent: It has merits (e.g., it reports state-of-the-art results, the idea is nice), but there are key weaknesses (e.g., it describes incremental work), and it can significantly benefit from another round of revision. However, I won't object to accepting it if my co-reviewers champion it.

**Paper Topic And Main Contributions:**

This paper proposes a new method for context-supervised pre-training for dense passage retrieval. The main idea is to use generated pseudo queries as context for pre-training. It can be combined with existing frameworks like coCondenser or CoT-MAE. The proposed method is evaluated on the MS-MARCO passage retrieval dataset and out-of-domain zero-shot BEIR benchmarks. Experimental results show that the proposed method is effective compared to baselines.


**Questions For The Authors:**

A. Why are some numbers on TREC DL for your own model implementations missing?


**Reasons To Accept:**

1. The paper is well written and easy to follow. The idea of using pseudo queries as context for pre-training is simple and clear.
2. Experiments on popular passage retrieval datasets show that the proposed method is effective and outperforms other competitive methods.


**Reasons To Reject:**

1. The technical contribution is not much. Generated pseudo queries have been widely used in the context of passage retrieval, including doc2query style sparse retrieval, zero-shot and few-shot dense retrieval. I acknowledge that this is the first paper to combine pseudo queries with coCondenser / CoT-MAE framework, but it is quite straightforward, and the overall contribution is limited.
2. The proposed method requires training a query generator with labeled data, while methods like coCondenser work with unlabeled data only. This will limit the applicability of the proposed method.


**Reproducibility:**

4: Could mostly reproduce the results, but there may be some variation because of sample variance or minor variations in their interpretation of the protocol or method.

**Reviewer Confidence:**

4: Quite sure. I tried to check the important points carefully. It's unlikely, though conceivable, that I missed something that should affect my ratings.

---

> ### Author Rebuttal · Authors · 2023-08-28
>
> Thank you very much for your insightful comments. We will answer your questions as follows.
>
> **Q1.** Why are some numbers on TREC DL for your own model implementations missing?
>
> **A1.** Sorry for the missing results. We accidentally ignored them in Table 1. We will fill in the below missing results on TREC DL immediately in our next release.
>
> | Model                                    | MS-MARCO |      |      | TREC DL 19 | TREC DL 20 |
> |------------------------------------------|----------|------|------|------------|------------|
> |                                          | MRR@10   | R@50 | R@1k | NDCG@10    | NDCG@10    |
> | coCondenser (120K) - retriever 1         | 37.0     | 86.0 | 98.5 | **68.2**       | 68.8       |
> | w/ query-as-context (120K) - retriever 1 | **37.4**     | **87.3** | **98.6** | 68.1       | **69.2**       |
> | CoT-MAE (50K) - retriever 1              | 37.2     | 85.7 | 98.2 | 65.7       | 66.5       |
> | w/ query-as-context (50K) - retriever 1  | **38.6**     | **87.7** | **98.6** | **67.7**       | **67.8**       |

---

### Official Review · Reviewer_nEeh · 2023-08-04

**Soundness:** 4

**Excitement:**

3: Ambivalent: It has merits (e.g., it reports state-of-the-art results, the idea is nice), but there are key weaknesses (e.g., it describes incremental work), and it can significantly benefit from another round of revision. However, I won't object to accepting it if my co-reviewers champion it.

**Missing References:**

- Yujing Wang, Yingyan Hou, Haonan Wang, Ziming Miao, Shibin Wu, Hao Sun, Qi Chen, Yuqing Xia, Chengmin Chi, Guoshuai Zhao, Zheng Liu, Xing Xie, Hao Allen Sun, Weiwei Deng, Qi Zhang, and Mao Yang. 2022. A neural corpus indexer for document retrieval. CoRR, abs/2206.02743.
- Shengyao Zhuang, Houxing Ren, Linjun Shou, Jian Pei, Ming Gong, Guido Zuccon, and Daxin Jiang. 2022. Bridging the gap between indexing and retrieval for differentiable search index with query generation. CoRR, abs/2206.10128
- Xing Wu, Guangyuan Ma, Peng Wang, Meng Lin, Zijia Lin, Fuzheng Zhang, Songlin Hu, CoT-MAE v2: Contextual Masked Auto-Encoder with Multi-view Modeling for Passage Retrieval, arXiv:2304.03158, 2023



**Paper Topic And Main Contributions:**

This paper proposes the use of a pair of query-passage for pretraining the dense retrieval, referred to as query-as-context pretraining, motivated by criticizing that a pair of passage-passage in the same document, which is used on the previous context-supervised pretraining, is weakly related or not relevant. The proposed query-as-context pretraining is applied to coCondenser and CoT-MAE, which shows improvements on the standard datasets.

**Questions For The Authors:**

1) Query generation has been widely used in DSI as below. Comparing the case of DSI, what distinguishable effects are expected by applying the query generation for pretraining dense retrieval?
- Yujing Wang, Yingyan Hou, Haonan Wang, Ziming Miao, Shibin Wu, Hao Sun, Qi Chen, Yuqing Xia, Chengmin Chi, Guoshuai Zhao, Zheng Liu, Xing Xie, Hao Allen Sun, Weiwei Deng, Qi Zhang, and Mao Yang. 2022. A neural corpus indexer for document retrieval. CoRR, abs/2206.02743.
- Shengyao Zhuang, Houxing Ren, Linjun Shou, Jian Pei, Ming Gong, Guido Zuccon, and Daxin Jiang. 2022. Bridging the gap between indexing and retrieval for differentiable search index with query generation. CoRR, abs/2206.10128

2) The last version of CoT-MAE is CoT-MAE-v2 that uses multi-view representation and decoding. Similar improvements could be made over CoT-MAE-v2 too?

3) In Table 3, the use of single query is most dominant, and increasing number of queries does not so make substantial difference. Can you check the diversity across generated queries? How to control the diversity of the generated queries?

4) In Table 3, it would help us to check the effect of single query when providing the case of the zero query number.


**Reasons To Accept:**

While query generation for pretraining has been widely used in DSI, the paper extensively applies the query generation on two existing models (i.e., coCondenser and CoT-MAE) in the dense passage retrieval, which makes meaningful and novel contribution in the literature. Different from the existing doc2query that focuses on document expansion, this work applies the query generation for the contrastive learning. In the experiment results, although the proposed method does not achieve the SOTA performance without improving CoT-MAE-v2, the performances are consistently improved on the standard datasets.

**Reasons To Reject:**

- The proposed query generation and its resulting pretraining is quite similar to the works introduced in DSI (i.e., NCI and DSI-QG), which makes the technical part less novel. Comparing to the existing NCI (or DSI-QG), the novelty and value of this work needs be clearly presented.

- The authors criticize that the use of a passage-passage pair in the same document is not optimal for pretraining the dense retrieval, and propose the query-as-context as an alternative approach. However, rather than the proposed alternative view, the proposed query-passage and the existing passage-passage pairs can be integrated in a complementary manner, based on the multi-task learning. Not all passage-passage pairs are problematic in pretraining, as some of the pairs would be relevant. The authors’ critic on the usefulness of the existing passage-passage pair is largely unvalidated. In particular, the experiments in Section 5.2, without mixing two types of pairs, the results of only passage-passage pair need to be presented. Experiments for the effect of combining two pretraining losses derived from passage-passage and passage-query pairs may be required.



**Reproducibility:**

4: Could mostly reproduce the results, but there may be some variation because of sample variance or minor variations in their interpretation of the protocol or method.

**Reviewer Confidence:**

4: Quite sure. I tried to check the important points carefully. It's unlikely, though conceivable, that I missed something that should affect my ratings.

**Typos Grammar Style And Presentation Improvements:**

In Section 3.1, in Appendix, it would be helpful to present the generated query samples for some passages.

---

> ### Author Rebuttal · Authors · 2023-08-28
>
> Thank you very much for your insightful comments. We will answer your questions as follows.
>
> **Q1.** Comparing the case of DSI, what distinguishable effects are expected by applying the query generation for pretraining dense retrieval?
>
> **A1.** DSI is a typical T5-based generation retrieval architecture. It receives queries as inputs and directly generates the document IDs. Thus DSI learns a direct mapping function from query to doc IDs, where queries partially come from query generation.
>
> In our method, query generation expands more accurate and fine-grained contexts compared to original context-based pre-training. It does not learn the mapping function like DSI but blends the expanded contexts for better passage representation pre-training.
>
> Thus we conclude the distinguishable effects as follows.
>
> 1) Different training objects. DSI aims at generation retrieval, while our method aims at context-based representation pre-training.
>
> 2) Different retrieval methods. DSI learns a direct mapping function from query to doc IDs, while our method retrieves passages with sentence representations.
>
> 3) Different query usages. DSI uses query generation directly in one fine-tuning stage (Or it does not distinguish pre-training or fine-tuning stage), while we use query generation in the pre-training stage, without changing the fine-tuning stage.
>
>
> **Q2.** Similar improvements could be made over CoT-MAE-v2 too?
>
> **A2.** Yes, query-as-context pre-training expands fine-grained contexts at the training data level, which can be applied
> to existing context-based pre-training methods like CoT-MAE-v2. We expect to observe similar improvements for CoT-MAE-v2, but due to the limited time for the response period, we will leave this exploration to our future works. Note that our paper is still self-contained as we focus on exploring context-based dense retrieval methods, while CoT-MAE-v2 uses both dense and sparse retrieval.
>
> **Q3.** In Table 3, the use of single query is most dominant, and increasing number of queries does not so make substantial difference. Can you check the diversity across generated queries? How to control the diversity of the generated queries?
>
> **A3.** The diversity of the generated queries is controlled through nucleus sampling by adjusting $top_p$ and $top_k$, referring to [1]. The current setting has already ensured diversity, and please kindly refer to the examples in Q5. We will open-source all our corpus with generated queries soon.
>
> Different numbers of queries per passage do not influence the results significantly. We hypothesize that this is due to the large amounts of pre-training corpus (3.2 million). Even if we only use one query per passage, the amount of queries is still enough for effective pre-training.
>
> [1] https://huggingface.co/blog/how-to-generate
>
> **Q4.** In Table 3, it would help us to check the effect of single query when providing the case of the zero query number.
>
> **A4.** Zero query means that we do not use query-as-context for pre-training. Thus zero query is the baseline result in Table 1. Please kindly refer to Table 1 for the baseline results coCondenser (120k) and CoT-MAE (50k). Moreover, we will add the results of zero query from Table 1 to Table 3 in our next version. Thanks for your question.
>
> **Q5.** About presenting the generated query samples for some passages.
>
> **A5.** Here is an example of passage and generated queries. Due to limited space for the author's response, we only present one example here. And we will present more as an Appendix of the paper in our next release.
>
> | Passage                                                                                                                                                                                                                                                                                                                               |
> |---------------------------------------------------------------------------------------------------------------------------------------------------------------------------------------------------------------------------------------------------------------------------------------------------------------------------------------|
> | The presence of communication amid scientific minds was equally important to the success of the Manhattan Project as scientific intellect was. The only cloud hanging over the impressive achievement of the atomic researchers and engineers is what their success truly meant; hundreds of thousands of innocent lives obliterated. |
>
> | Queries                                                                                              |
> |------------------------------------------------------------------------------------------------------|
> | What were the biggest contributions of the manhattan project?                                        |
> | What was the importance of communication between scientists and each other for the mandela project?  |
> | Why was the madamson project important?                                                              |
> | How did the manhattan project work?                                                                  |
> | Why was the mandela project important to the history of science?                                     |
> | Why was the new york project important?                                                              |
> | Why the manhattan project was so important to society?                                               |
> | Why was the manhattan project important to the scientific world?                                     |
> | Why the manhattan project was so important?                                                          |
> | What did the manhattan project have in common with the atomic bomb?                                  |

---

### Official Review · Reviewer_DMcN · 2023-08-05

**Soundness:** 3

**Excitement:**

3: Ambivalent: It has merits (e.g., it reports state-of-the-art results, the idea is nice), but there are key weaknesses (e.g., it describes incremental work), and it can significantly benefit from another round of revision. However, I won't object to accepting it if my co-reviewers champion it.

**Paper Topic And Main Contributions:**

This paper proposes a pre-training technique designed to address the problem of weakly correlated pairs of passages within the same document. The proposed method assumes that a query derived from a passage is likely to be more relevant to that passage, thereby forming a passage-query pair.
The experimental results demonstrate that the method yields substantial improvements, thereby showcasing its effectiveness and efficiency.








**Questions For The Authors:**

1. How were the values of topp and topk determined for T5 fine-tuning?
2. The performance could vary depending on how the T5 model is pre-trained. Additionally, what about exploring the use of other generation models besides T5?





**Reasons To Accept:**

Strongness
1. They proposed a simple context-supervised pretraining technique by utilizing query-passage pairs.
2. The results of the experiment show that the method delivers significant enhancements.
3. They also examined the impact of the number of generated queries and the mixed context.

**Reasons To Reject:**

Weakness
1. The criteria for dividing passages need to be more accurate. Instead of simply dividing by token length, shouldn't the division be based on semantic units?


**Reproducibility:**

4: Could mostly reproduce the results, but there may be some variation because of sample variance or minor variations in their interpretation of the protocol or method.

**Reviewer Confidence:**

3: Pretty sure, but there's a chance I missed something. Although I have a good feel for this area in general, I did not carefully check the paper's details, e.g., the math, experimental design, or novelty.

---

> ### Author Rebuttal · Authors · 2023-08-28
>
> Thank you very much for your insightful comments. We will answer your questions as follows.
>
> **Q1.** The criteria for dividing passages need to be more accurate. Instead of simply dividing by token length, shouldn't the division be based on semantic units?
>
> **A1.** Dividing by semantic units is an intriguing option for fine-grained context-supervised pre-training. However, some aspects may prevent such usage.
>
> 1) Processing efficiency: Dividing pre-training corpus as semantic-related spans is not an easy task, especially when dealing with millions of documents from Wikipedia or Web Pages like MS-MARCO. Dividing by token length shows better processing efficiency.
>
> 2) Effectiveness: The boundary of semantic units may not be very clear sometimes. Existing semantic processing tools may bring bias to the processed corpus.
>
> Thus for processing efficiency and effectiveness, a popular option for dividing passages in unsupervised context-based pre-training systems [1,2] is still directly cutting them by token length. Thanks for your insightful advice on semantic units, we will seek to solve these issues in future works.
>
> [1] Unsupervised Corpus Aware Language Model Pre-training for Dense Passage Retrieval. ACL (1) 2022: 2843-2853
>
> [2] Towards Unsupervised Dense Information Retrieval with Contrastive Learning. CoRR abs/2112.09118 (2021)
>
> **Q2.** How were the values of top-p and top-k determined for T5 fine-tuning?
>
> **A2.** We basically follow the guidance on Huggingface [1] for nucleus sampling. We tested several combinations of top-p and top-k in preliminary experiments and found that $top_p=0.95$ and $top_k=25$ bring diverse queries and a stable generation process.
>
> [1] https://huggingface.co/blog/how-to-generate
>
> **Q3.** The performance could vary depending on how the T5 model is pre-trained. Additionally, what about exploring the use of other generation models besides T5?
>
> **A3.** We use the widely adopted generative T5 model as a query expansion model [1] for reproducibility. We will also explore using more generation models besides T5 in our future works, such as the recent powerful ChatGPT-like large language models. Thanks for your advice.
>
> [1] From doc2query to docTTTTTquery[J]. Online preprint, 2019, 6: 2.

---

### Meta-Review · Area_Chair_xXPp · 2023-09-19

**Recommendation:** 4

**Metareview:**

This paper proposes an approach to generate queries from a passage, and use them to pre-train a dense retriever. The approach is an alternative to using passage-passage pairs for pre-training. The experiments demonstrate the positive effects using such a pre-training approach.

The idea is interesting, although it is a straightforward extension to the exiting work. The experiments successfully demonstrate that this is a reasonable way to create training datasets for dense retriever.

The authors have answered most of the questions raised by the reviewers in their rebuttal.

---

### Decision · Program_Chairs · 2023-10-07

**Decision:**

Accept-Main

**Comment:**

This paper proposes an approach to generate queries from a passage, and use them to pre-train a dense retriever. The approach is an alternative to using passage-passage pairs for pre-training. The experiments demonstrate the positive effects using such a pre-training approach.

The idea is interesting, although it is a straightforward extension to the exiting work. The experiments successfully demonstrate that this is a reasonable way to create training datasets for dense retriever.

The authors have answered most of the questions raised by the reviewers in their rebuttal.